# Yolo-Light: Remote Straw-Burning Smoke Detection Based on Depthwise Separable Convolution and Channel Attention Mechanisms

Rui Hong [1], Xiujuan Wang [2,*], Yong Fang [1], Hao Wang [2], Chengpeng Wang [3] and Huanqin Wang [3,4,*]

[1] National Engineering Lab of Special Display Technology, State Key Lab of Advanced Display Technology, Academy of Opto-Electronic Technology, Hefei University of Technology, Hefei, 230009, China

[2] School of Microelectronics, Hefei University of Technology, Hefei 230009, China

[3] State Key Laboratory of Transducer Technology, Institute of Intelligent Machines, Hefei Institutes of Physical Science, Chinese Academy of Sciences, Hefei 230031, China

[4] Department of Automation, University of Science and Technology of China, Hefei 230026, China

* Correspondence: xjwang2022@hfut.edu.cn (X.W.); hqwang@iim.ac.cn (H.W.)

**Abstract:** Straw burning is a long-term environmental problem in China's agricultural production. At present, China relies mainly on satellite remote sensing positioning and manual patrol to detect straw burning, which are inefficient. Due to the development of machine learning, target detection technology can be used for the detection of straw burning, but the current research does not take into account the various scenarios of straw burning and the deployment of object detection models. Therefore, a lightweight network based on depthwise separable convolution and channel attention mechanisms is proposed to detect straw-burning smoke at a remote distance. Various regional and crop-burning smoke datasets were collected to make the algorithm more robust. The lightweight network was applied to automatically identify and detect straw-burning smoke in surveillance videos. The experiment showed that the amount of light network parameter was only 4.76 M, and the calculation performance was only 11.2 Gflops. For the intelligent detection of straw-burning smoke, performance verification accuracy was improved by 2.4% compared with Yolov5s. Meanwhile, the detection speed on the embedded Jetson Xavier NX device can reach 28.65 FPS, which is 24.67% better than the Yolov5s. This study proposes a lightweight target detection network, providing a possible method for developing low-cost, rapid straw-burning smoke detection equipment.

**Keywords:** straw-burning smoke detection; lightweight network; depthwise separable convolution; channel attention

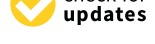



## 1. Introduction

China is a major agricultural country, and the agricultural economy accounts for a significant proportion of China's economic structure, which has a significant impact on the long-term and stable development of the social economy. However, there is a long-term environmental problem in China's agricultural production, which is straw burning [1]. Research shows that the total amount of pollutants generated by straw burning in Hebei Province accounts for 21% of the total air pollutants [2]. The government has been aware of the hazards of straw burning since 2016, and in order to address this issue, corresponding policies have been gradually introduced to prohibit straw burning on-site. Relevant research shows that burning straw not only pollutes the environment but also harms the health of residents nearby [3,4]. At present, law enforcement departments rely mainly on satellite remote sensing positioning and manual patrol to detect straw burning. Remote sensing recognition can only detect a certain range of straw burning, which will cause many omissions. Artificial methods are inefficient and harmful to the health of law enforcement personnel [5,6]. Due to the area of the straw-burning flame being small, it is difficult to observe the flame of straw burning at a long distance. Since the large amount of smoke

generated by straw burning can be monitored at long distances, it is feasible to judge straw burning by monitoring smoke. Thus, an urgent need to develop a high-precision, non-contact, rapid, remote detection method for straw-burning smoke exists. Using target detection technology to build a straw-burning smoke target detection system can help to meet the above requirements and is highly suitable for use by law enforcement agencies to detect straw-burning smoke.

The development of deep learning opens a new route to straw-burning smoke detection. Researchers have successfully applied deep learning for smoke detection and have obtained higher than traditional detection accuracy and speed using the target detection technology to build the straw-burning smoke target detection system. This system can meet the above requirements and is suitable for enforcing straw-burning smoke. Currently, the commonly used target detection algorithms include Faster R-CNN (Region-Convolutional Neural Network) [7], SSD (Single Shot MultiBox Detector) [8], and Yolo (You Only Live Once) [9–11]. In terms of straw-burning smoke detection, Wang et al. [12] proposed a smoke-detection method based on an improved frame difference method and Faster R-CNN. For smoke detection, it first uses the improved frame difference method to extract candidate regions, then it uses the Faster R-CNN model for smoke detection. The model in this paper uses Faster R-CNN with slow reasoning speed and the inter-frame difference method to filter data, which cannot meet the needs of real-time monitoring by law enforcement personnel. Anhui Baolong Environmental Protection Technology [13] have also attempted to add the straw-burning smoke detection function to their remote sensing monitoring equipment; however, within the actual applications, the smoke-detection speed of this equipment is low.

Liu et al. [14] proposed using the improved Yolov5s algorithm to detect smoke in Sentinel-2 images captured through remote sensing. A convolutional block attention module was added to the original model to improve detection accuracy. However, this algorithm is used for remote sensing monitoring. In practice, monitoring the small area of straw burning is impossible. Mukhiddinov et al. [15] and others conducted studies by placing a small arithmetic platform on an unmanned aerial vehicle, and used Yolov5 for real-time reasoning to complete outdoor wildfire remote monitoring. In the article, the authors optimize anchors using K-mean++ technology to reduce classification errors, and focus on small- and medium-sized wildfire smoke areas with a spatial pyramid structure. The results of the paper show that the improved Yolov5 recognition rate with the dataset is 73.6%. The experiment verifies that Yolov5 can be successfully used for the detection of straw-burning smoke. However, the modification does not account for whether the speed and accuracy of the model can be further optimized, and more research is needed.

The current research does not consider the large-scale monitoring of straw-burning smoke. Firstly, when observing from a distance, the area of smoke is relatively small, and targeted optimization of the algorithm is needed. At the same time, the morphology of smoke observed at different distances is different, so it is necessary to collect smoke data from different distances. Secondly, the background color of the same location varies in different seasons, such as green fields in spring and yellow fields in autumn. Therefore, it is necessary to collect smoke data from straw burning across different seasons. Finally, existing research has not considered optimizing the speed and accuracy of algorithms. When the hardware requirements of the algorithm are too high, and the algorithm is applied in the wild, it is not conducive to deployment. This article will research these issues.

Based on Yolov5s [16], this study proposes a target detection algorithm with a small model size and a fast detection speed [17]. A lightweight network is very suitable for mobile and embedded devices [18]. Based on the Yolov5s network, the network's backbone is improved, and the depthwise separable convolution [19,20] and ECA (Efficient Channel Attention) mechanism [21] are introduced. We call the improved network Yolo-Light. Datasets are extremely important for machine learning. Currently, there are no public datasets of straw-burning smoke, so datasets of straw-burning smoke in different seasons, locations, and crops were collected in this study.

Yolo-Light was trained using the straw-burning smoke target dataset and deployed on the Jetson Xavier NX embedded platform. The straw-burning smoke target dataset was used to train Yolo-Light and Yolov5s, and the model parameters and performance were compared. Moreover, the actual detection ability of Yolo-Light on the embedded Jetson Xavier NX device was verified. Compared with Yolov5s, Yolo-Light has fewer parameters and calculations, and can achieve faster detection speeds on embedded devices.

## 2. Materials and Methods

### 2.1. Basic Principles of the Yolov5 Target Detection Model

Due to the need to improve the detection speed, the Yolo series cancels the pooling and full connection layers, and adopts the method of using only the volume layer. Since the pooling layer is withdrawn, the down sampling of the Yolo series is realized through the stripe = 2 of the convolution layer. To detect objects of different sizes, the Yolo series uses three scales of different sizes to improve the recognition rates of small targets.

Yolov5 has a more robust performance and a smaller volume than previous versions. Four versions of the Yolov5 target detection network exist, and the model sizes are successively increased to Yolov5s, Yolov5m, Yolov5l, and Yolov5x. The Yolov5s network has the lowest depth and feature-map width within the Yolov5 series; it also has the fastest detection speed. Moreover, it is very suitable for embedded devices with small calculations. The other three have been continuously deepened and improved upon this basis. The network structure diagram is shown in Figure 1.

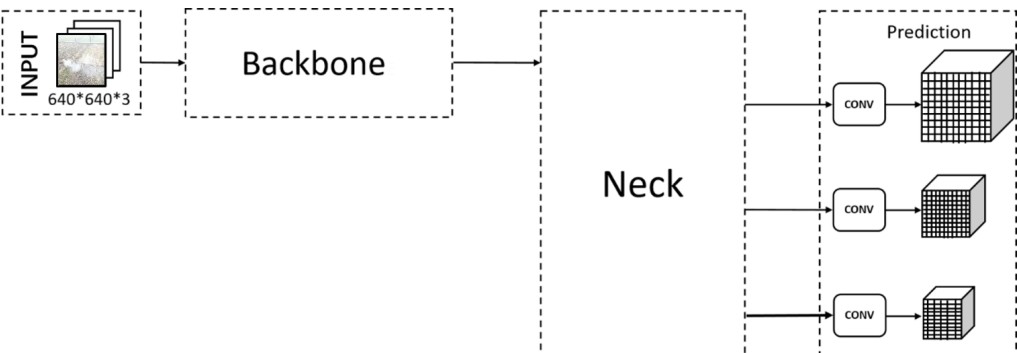

**Figure 1.** Network structure of Yolov5.

As can be seen from the network structure diagram, Yolov5 is divided into four parts: input, backbone, neck, and prediction. The Yolov5 series primarily controls the depth of the network by controlling the number of residual components in BottleneckCSP. The comparison of the number of residual features in BottleneckCSP is shown in Table 1.

**Table 1.** Network structure comparison of Yolov5.

| Section | Structure | Yolov5s | Yolov5m | Yolov5l | Yolov5x |
|---|---|---|---|---|---|
| backbone | BottleneckCSP1 | 1 | 3 | 3 | 4 |
| | BottleneckCSP1 | 3 | 6 | 9 | 12 |
| | BottleneckCSP1 | 3 | 6 | 9 | 12 |
| | BottleneckCSP2 | 1 | 2 | 3 | 12 |
| head | BottleneckCSP2 | 1 | 2 | 3 | 4 |
| | BottleneckCSP2 | 1 | 2 | 3 | 4 |
| | BottleneckCSP2 | 1 | 2 | 3 | 4 |
| | BottleneckCSP2 | 1 | 2 | 3 | 4 |

Since the backbone of Yolov5s is considerably large, it needs to be optimized to compress the model volume and to increase the detection speed. This chapter proposes a lightweight network based on Yolov5s, which can perform well on the embedded platform.

Compared with Yolov5s, its main improvements are its ordinary convolution to depthwise separable convolution and the introduction of the ECA mechanism.

### 2.2. Lightweight and High-Precision Target Detection Network

#### 2.2.1. Depthwise Separable Convolution

Compared with traditional convolution, the most significant advantage of depthwise separable convolution [15] is the reduction of the parameters. For a feature map $H \times w \times c_1$, after a standard convolution (assuming the size of the convolution kernel is $h \times w \times c_1$ and that the convolution has the padding operation), the output size is $H \times w \times c_2$; then the parameter of standard convolution is shown in (1).

$$P_{conv} = (h \times w \times c_1) \times c_2 \tag{1}$$

Depthwise separable convolution performs a spatial convolution while keeping the channel independent, and then performs the depth convolution operation. Depthwise separable convolution is a combination of depthwise convolution and pointwise convolution. Depthwise convolution is responsible for filtering. Its convolution kernel size is $h \times w \times 1$ and it has a total of $c_1$, which acts on each channel. Pointwise convolution is accountable for the conversion channel, and the convolution kernel size is $1 \times 1 \times c_1$. There are $c_2$ in total, which act on the output feature map of depthwise convolution. The parameter of depthwise convolution is shown in (2), and the parameter of pointwise convolution is shown in (3).

$$P_{Depthwise} = (h \times w \times 1) \times c_1 \tag{2}$$

$$P_{Pointwise} = (1 \times 1 \times c_1) \times c_2 \tag{3}$$

The parameters of depthwise separable convolution are shown in (4).

$$P_{Depthwiseseparable} = P_{Depthwise} + P_{Pointwise} = h \times w \times c_1 + c_1 \times c_2 \tag{4}$$

The comparison between depthwise separable convolution and standard convolution is shown in (5).

$$\frac{P_{Depthwiseseparable}}{P_{conv}} = \frac{h \times w \times c_1 + c_1 \times c_2}{h \times w \times 1 \times c_1 \times c_2} = \frac{1}{c_2} + \frac{1}{(h \times w)^2} \tag{5}$$

Depthwise separable convolution effectively decomposes the traditional convolution by separating the spatial filtering from the feature generation mechanism and reducing the amount of calculation by eight to nine times, compared with the traditional convolution [20]. The structure of traditional convolution (a) and depthwise separable convolution is shown in the Figure 2.

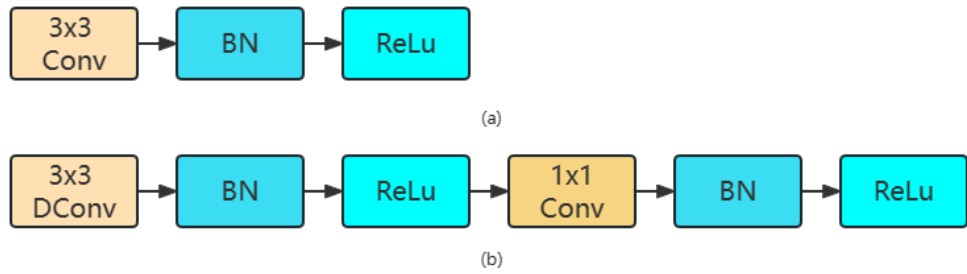

**Figure 2.** Traditional convolution (**a**) and depthwise separable convolution (**b**).

### 2.2.2. ECA Attention Mechanism

ECA is an effective channel attention module, which only adds a few parameters, but this can lead to a clear performance gain.

The attention mechanism allows the network to selectively enhance the features with a large amount of information, so that these features can be fully utilized in subsequent processing, and then useless features are suppressed. Taking traditional convolution as an example, the convolution kernel is $V = [v_1, v_2, v_3, \cdots, v_c]$, where $v_c$ represents the $c$-th convolution kernel. For input $X$, the output is $U = [u_1, u_2, u_3, \cdots, u_c]$.

$$u_c = v_c * X = \sum_{s=1}^{c} v_c^s * x^s \tag{6}$$

where $*$ in (6) represents a convolution operation, $v_c^s$ represents a convolution kernel, its input is a spatial feature on a channel, and it learns a spatial relationship. However, due to a summation operation performed to obtain the convolution results of each channel, the channel feature relationship is mixed with the spatial relationship learned by the convolution kernel. The purpose of the attention mechanism operation is to extract this mixed operation, so that the model directly learns the characteristic relationship of each channel.

The ECA net mainly improves the SE (Squeeze and Excitation) module [22], and proposes a local cross-channel interaction strategy (ECA module) without dimensionality reduction and a method of adaptively selecting the size of the one-dimensional convolution kernel, so as to improve the performance. The structure of SE Module is shown in the Figure 3.

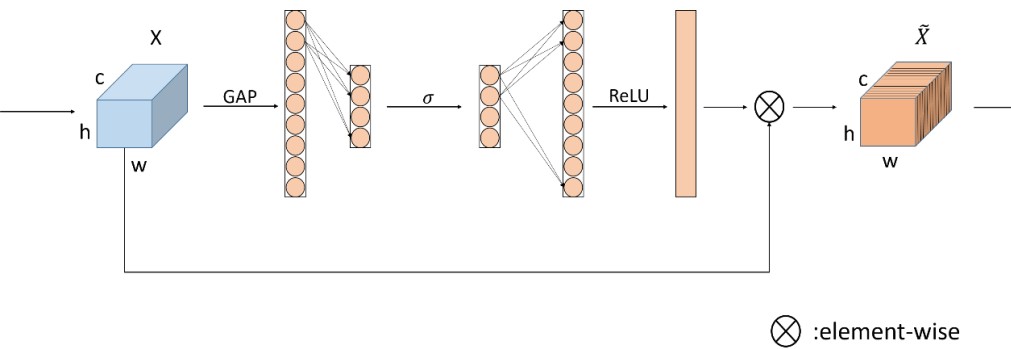

**Figure 3.** SE Module.

As shown in Figure 4, after performing channel-level global average pooling without reducing the dimension, the local cross-channel interaction information is captured by considering each channel and its k-nearest neighbors. Cross-channel interaction information is captured via rapid one-dimensional convolution with size K.

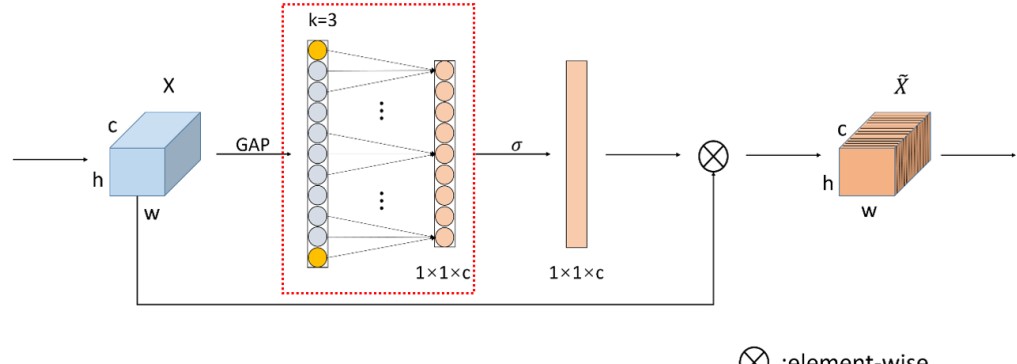

**Figure 4.** ECA Module.

The ECA module is developed on the SE module. Two fully connected layers are used in the SE module to obtain the correlation between each channel. The two fully connected layers perform dimensionality reduction operations on the global feature compression obtained through global average pooling, which affects the acquisition of the correlation between each channel. The ECA module has improved this.

The ECA increases information sharing and interconnection between convolution channels, avoids dimensionality loss and information loss during information sharing, and significantly reduces model complexity while maintaining performance.

$$
\begin{bmatrix}
w^{1,1} & \cdots & w^{1,k} & 0 & 0 & \cdots & \cdots & 0 \\
0 & w^{2,2} & \cdots & w^{2,k+1} & 0 & \cdots & \cdots & 0 \\
\vdots & \vdots & \vdots & \vdots & \ddots & \vdots & \vdots & \vdots \\
0 & \cdots & 0 & 0 & \cdots & w^{C,C+1} & \cdots & w^{C,C}
\end{bmatrix}
\tag{7}
$$

As shown in (7), the ECA module uses the matrix $W_k$ to learn channel attention. $W_k$ has a total of $k * C$ parameters, which avoids complete independence between different groups. The weight of the i-th channel is calculated via the correlation between the *i*-channel and its $k$ adjacent channels. The calculation formula is shown as (8).

$$
\omega_i = \sigma(\sum_{j=1}^{k} w_i^j y_i^j), y_i^j \in \omega_i^k
\tag{8}
$$

where $\omega_i$ represents the learned channel attention, $y_i$ represents the output of the *i*-th channel after global average pooling, and $\omega_i$ represents the set of k adjacent channels of $y_i$. All channels can share weight information to reduce the amount of parameters and further improve performance.

One-dimensional convolution with a convolution kernel size of $k$ is used to achieve information interaction between channels.

$$
\omega = \sigma(C1D_k(y))
\tag{9}
$$

In (9), C1D represents one-dimensional convolution, which only involves $k$ parameter information, and it generally has a value of $k = 3$. This method of capturing cross-channel information interaction can ensure performance and model efficiency.

### 2.2.3. The Overall Structure of the Yolo-Light Network

The improved network structure is shown in Figures 5 and 6. The backbone of the network uses two bottleneck structures. The ECA module is added to Bottleneck1 but is not used in Bottleneck2. In both the bottleneck structures, depth-separable convolutions are used instead of traditional convolutions to reduce the number of parameters.

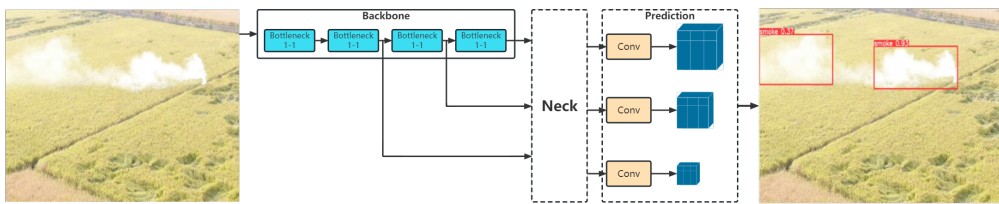

**Figure 5.** Network structure of improved model.

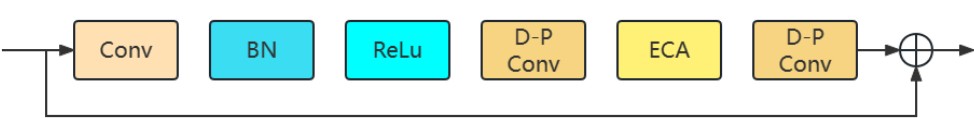

**Figure 6.** Network structure of Bottleneck.

Compared with Yolov5s, Yolo-Light reduces the number of network layers and parameters, and improves speed and accuracy.

### 2.3. Collect and Build Dataset

As the current research on straw-burning smoke is in its infancy, we need a relevant dataset. The open-source smoke datasets are mainly fire accident smoke datasets and are unsuitable for monitoring straw-burning smoke. We therefore required a dataset for the target detection of straw incineration smoke. As shown in Figure 7a, a 2-megapixel camera with a 10–360 mm zoom lens was designed to capture a straw-burning smoke video at a distance of 3–5 km. The camera model is DS-2CD5027EFWD-A, manufactured by Hikvision. It has a 1/8″ COMS (complementary metal-oxide semiconductor) sensor, a dynamic range of 120 db, supports H.265, and has a resolution of 1920 × 1080. The power consumption of the camera is only 6 W, and the operating temperature is −30 °C to −60 °C, making it suitable for deployment in the wild. In order to monitor long-distance straw-burning smoke, we equipped the camera with a 10–360 mm zoom lens. The model of the lens is YM36 × 10MAPRF, manufactured by YAMAKO. The lens has a temperature compensation function, with a working temperature of −40 °C to −70 °C. The combination of camera and lens can achieve the collection and monitoring of straw-burning smoke within a range of 3–5 km. As shown in Figure 7b, an unmanned aerial vehicle (UAV) with a camera is used to collect smoke from straw burning at close range. The UAV model is DJI MAVIC PRO, manufactured by DJI, with a 12.35-megapixel camera. The camera resolution is 3840 × 2160 with a 26 mm lens. The UAV is used to capture straw-burning smoke footage at a distance of 1–2 km.

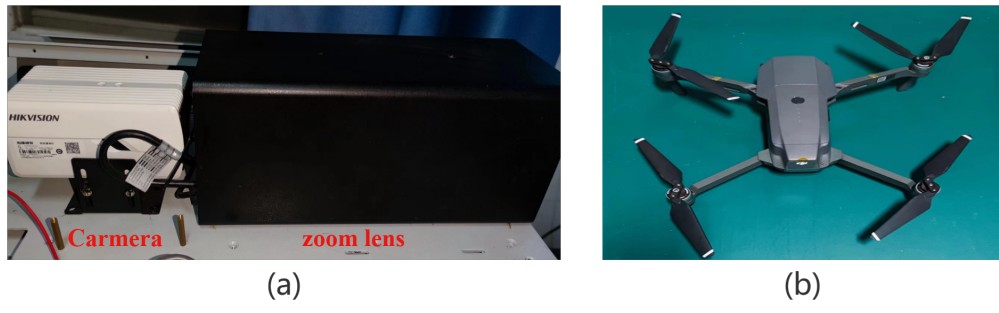

(a)                                          (b)

**Figure 7.** Zoom camera (**a**) and unmanned aerial vehicle (UAV) (**b**).

To enrich the dataset, we collected straw-burning smoke data in the summer under the background of immature crops and straw-burning smoke data in the autumn after the crops were mature and harvested. Due to the different colors of smoke burned by different crop straws, the smoke from wheat, maize, and rice straws were collected in this study. At the same time, considering the different backgrounds in different geographic environments, the experiment of collecting straw smoke data was arranged in different locations, including Hefei (Anhui Province), Luan (Anhui Province), Huainan (Anhui Province), Shangqiu (Henan Province), and Shenyang (Liaoning Province). Some pictures of training dataset and test dataset in different locations is shown in the Figure 8.

This study collected smoke videos for about 5 h, and selected 10,000 different smoke pictures as the training dataset and 1000 different pictures not in the training dataset as the test dataset. The number of images in the dataset is shown in Table 2. The different backgrounds of the collected smoke datasets include seasons (spring and autumn), locations (southern and northern regions of China), terrain (plains and hills), crops (rice, wheat, and corn), weather (sunny and cloudy), time (morning, afternoon, and dusk), and distance.

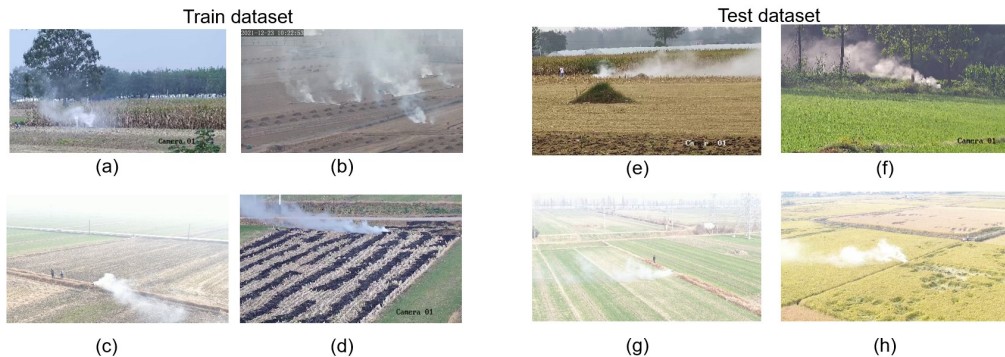

**Figure 8.** Some pictures of training dataset and test dataset in different locations. (**a**,**e**) Corn fields in Henan Province; (**b**) paddy fields in Heilongjiang Province; (**c**,**g**) wheat fields in Anhui Province; (**d**,**f**,**h**) paddy fields in Anhui Province.

**Table 2.** Datasets for training and testing.

|  | Training | Test |
|---|---|---|
| Paddy field | 6000 | 600 |
| Corn field | 2000 | 200 |
| Wheat field | 2000 | 200 |
| Total | 10,000 | 1000 |

### 2.4. Quality Control

Firstly, in order to control the quality of data collection, this study collected three scenarios of straw-burning smoke, including paddy fields, corn fields, and wheat fields. Secondly, due to changes in background color caused by seasonal changes, this study collected data from spring and autumn. Thirdly, considering the complexity of China's region, this study collected data on straw burning smoke in southern and northern regions of China, including plains and hills. Fourthly, in order to handle backgrounds with different brightness levels, the weather conditions for collecting data in this study included sunny and cloudy days, and the collection times included morning, afternoon, and dusk. Finally, in order to control the impact of the size of the straw-burning smoke area, this study collected smoke images from both near and far distances during data collection. These measures ensured the balance and diversity of the data.

During the modeling process, this study used the following parameters to train all models: epochs = 300, img-size = [640, 640], batch-size = 64. Epochs represented the number of iterations of the model, img-size represented the image size of the input model, and batch-size represented the number of images trained for each input model. After the training, the model with the highest mAP(mean Average Precision) value was selected as the final output of modeling.

### 2.5. Experimental Platform

The experimental platform in this paper is shown in Table 3. The CPU used to train the model is an Intel Core i9-10900X, which is a high-performance CPU with ten cores and twenty threads, and a frequency of 3.7 GHz. The GPU used is NVIDIA TITAN RTX, which has a 24 GB GDDR6 frame buffer, a boost clock of 1770 MHz, and 4608 CUDA cores. The excellent performances of CPU and GPU ensured the stability of the training model.

**Table 3.** Train model environment.

| System | Ubuntu18.04 |
|---|---|
| CPU | Intel Core i9-10900X |
| GPU | 2 × TITAN RTX |
| Hardware acceleration | Ubuntu18.04; CUDA10.1; Pytorch1.11; opencv4.6 |

## 3. Results

### 3.1. Model Size

The self-built dataset was used to train Faster R-CNN, Yolov5s, and Yolo-Light under the above-mentioned experimental platform. The model with the highest mAP was updated and retained during training. The final model parameters are shown in Table 4.

**Table 4.** Comparison of network model parameters.

| Model | Params | Flops | Model Size |
|---|---|---|---|
| Faster R-CNN | 28.3 M | 213 G | 114.2 MB |
| Yolov5s | 7.01 M | 15.8 G | 14.4 MB |
| Yolov5s+ECA | 7.22 M | 16.1 G | 14.8 MB |
| Yolov5s+Dconv | 4.56 M | 10.9 G | 9.5 MB |
| Yolo-Light | 4.76 M | 11.2 G | 9.9 MB |

Table 4 shows that Yolo-Light has the lowest parameters and calculations, compared with Yolov5s and Faster R-CNN. The model volume is only 9.9 MB. Compared with the Yolov5s model, the volume is reduced by 31.3%, the parameter volume is reduced by 31.1%, and the calculation amount is reduced by 29.11% (the test input is [1, 3, 640, 640]). The Faster R-CNN model has the largest volume, at 114.2 MB, which is 11.54 times that of Yolo-Light; the calculation performance is 19.01 times that of Yolo-Light. The above results indicate that compared to Fast R-CNN, the one-stage target-recognition algorithm has a significant advantage in parameter quantity, which is beneficial for deployment on embedded platforms. After adding ECA, the parameter count of Yolov5s increased slightly, while the parameter count of Yolo Light with Dconv decreased even further.

### 3.2. Comparison Experiment

In order to verify the effect of the ECA attention mechanism, this study designed some ablation experiments. As the basis of ECA's attention mechanism, the SE attention mechanism is added to Yolov5s for comparison. In theory, the smoke target is mainly a special color feature, and so the spatial attention mechanism has little impact on the network. In order to verify this theory, the convolutional block attention module (CBAM) attention mechanism [23], which integrates the spatial attention mechanism and the channel attention mechanism, is also used as the contrasting network in this study. Table 5 shows the results of the comparison.

**Table 5.** Ablation experiment.

| Model | Params | Flops | mAP.5 |
|---|---|---|---|
| Yolov5s | 7.01 M | 15.8 G | 0.709 |
| Yolov5s+CBAM | 7.22 M | 16.1 G | 0.695 |
| Yolov5s+SE | 7.21 M | 16.1 G | 0.725 |
| Yolov5s+ECA | 7.21 M | 16.1 G | 0.732 |
| Yolov5s+Dconv | 4.56 M | 10.9 G | 0.708 |
| Yolo-Light | 4.76 M | 11.2 G | 0.750 |

The results show that channel attention significantly improved the detection of straw-burning smoke, especially the ECA attention mechanism. In contrast, the CBAM attention mechanism, which integrates the spatial attention mechanism, has a negative effect on the network. The reason for this situation is that the smoke in the dataset does not appear at a fixed position in the image, and so the effect of CBAM will be worse. Meanwhile, due to the concentration of smoke colors in white, gray, and black, and the input image of the model being an RGB (red, green, blue) channel, channel attention is more suitable for detecting straw-burning smoke. The results show that the depthwise separable convolution can greatly reduce the computational load, but it does not reduce the network performance.

The calculation method of Dconv is point convolution and channel convolution, so for smoke images with an obvious channel information input, parameter reduction can be achieved without reducing accuracy. Finally, Yolo-Light, which combines the ECA module and deep separable convolution, reduces the number of model parameters and increases the mAP.5 value by 4.1% compared with Yolov5s.

### 3.3. Hardware Deployment

To analyze the reasoning speed of the improved model on the embedded platform, this article deploys the network to Jetson Xavier NX.

Jetson Xavier NX is an embedded device developed by NVIDIA. It uses a 64-bit quad-core ARM A57 processor with a working frequency of 1.4 GHz, has 384 CUDA cores, can provide 845 GFLOPS floating-point computing capabilities, consumes only 10 W of power, and the price is only USD 399. The operating environment of JetPack4.6, developed by NVIDIA, provides a complete desktop Linux environment and supports the CUDA Toolkit and cuDNN.

The network is deployed to Jetson Xavier NX and is tested on the test set; the results are shown in Table 6.

**Table 6.** Performance comparison of models on Jetson Xavier NX.

| Model | Inference Time | FPS on Jetson |
|---|---|---|
| Yolov5s | 43.3 ms | 22.57 |
| Yolov5s+ECA | 44.7 ms | 22.37 |
| Yolov5s+Dconv | 34.2 ms | 29.24 |
| Yolo-Light | 34.9 ms | 28.65 |

Due to the lower computational complexity of Dconv compared to traditional convolution, Yolo Light's inference speed improved by 27% compared to Yolov5s, and the frame rate for NX reached 28.65 FPS (frames per second). Compared to the improvement in the accuracy of ECA, the increase in the algorithm parameter quantity is very low, and the inference speed difference between Yolo-Light and Yolov5s+Dconv is only 0.2 ms. This result indicates that Yolo-Light can easily implement real-time reasoning on embedded platforms. At the same time, a miniaturized instrument based on Yolo-Light can be developed to monitor the smoke from straw burning.

### 3.4. Performance Analysis

To further verify the network's detection performance for straw-burning smoke, we collected 1000 straw-burning smoke pictures in different backgrounds to test the performance of the networks. Different backgrounds include paddy fields, corn fields, and wheat fields in spring and autumn. At the same time, in order to verify the robustness of the algorithm, some unfamiliar backgrounds were added, including green lawns and low visibility fields. The performance test index is the straw-burning smoke target-recognition rate, and its calculation formula is shown in (10).

$$Recognition\ rate = \frac{The\ number\ of\ correct\ picture}{Total\ number\ of\ pictures} \times 100 \qquad (10)$$

The number of correct pictures function shows the number of pictures correctly recognized by the model (no false and no missed detections).

In Figure 9, the scene shows a paddy field in Luan in spring. The background has a cement floor similar to the color of the straw-burning smoke and the field is green. The color contrast of the picture is high, which makes it easy to encounter errors. From the test results, it can be seen that Yolov5s and Yolov5s+DConv missed the smoke in the diagram. Yolov5s+ECA encountered a mistake in identifying roads as smog; Yolo-Light did not show any mis-detection and correctly detected smoke.

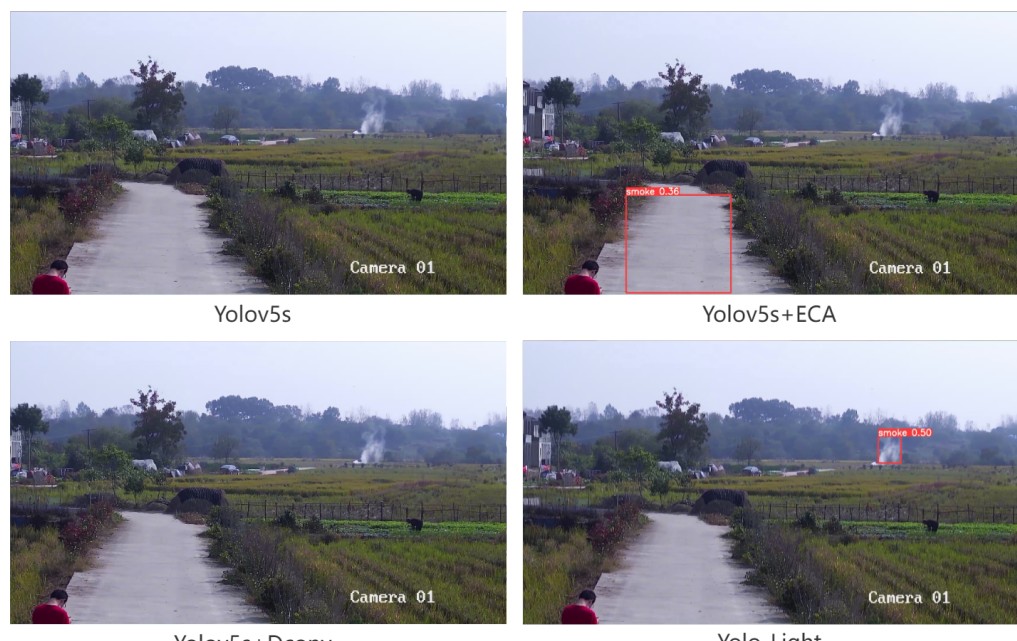

**Figure 9.** Detection results of different algorithms in spring.

In Figure 10, the scene shows a paddy field in Luan in autumn. There is no straw burning in this scene. The main picture is yellow and displays mature rice. There is a reflective pond in the paddy field. Under the same background, the algorithm is prone to misinterpret the reflections of the pond as smoke. From the test results, Yolov5s mistakenly identified the pond as smoke; Yolov5s+ECA, Yolov5s+DConv, and Yolo-Light were not mistaken.

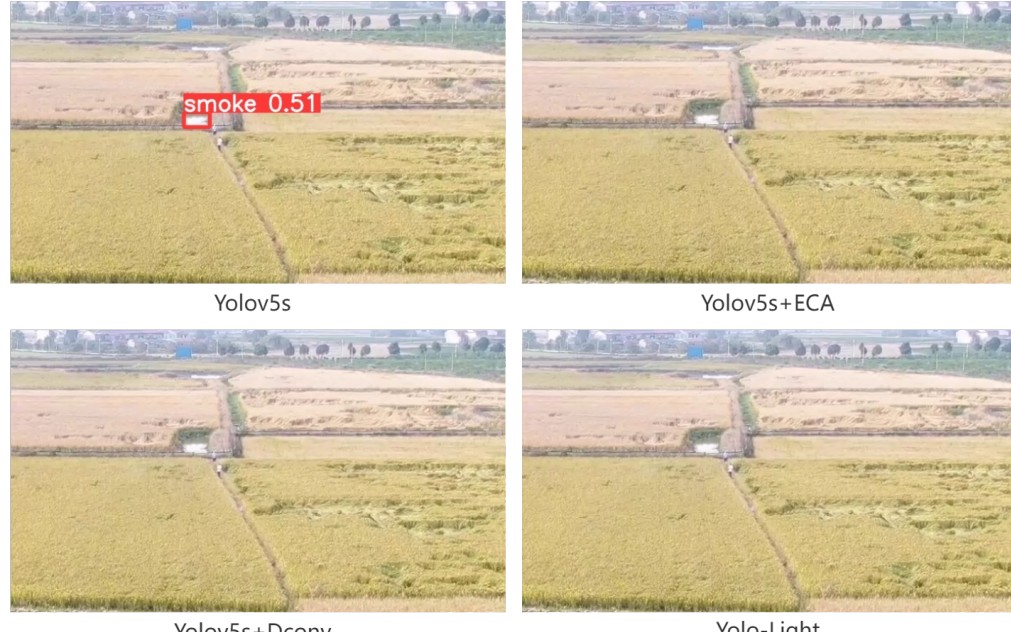

**Figure 10.** Detection results of different algorithms in autumn.

The scene in Figure 11 shows a green lawn smoke-detection scenario, which is not within the test range of the training set. The scene can be used to test the robustness of the smoke detection algorithm, which proves that the algorithm still has a stable detection ability in unfamiliar scenes. From the test results, we can see that both Yolov5s and

Yolov5s+DConv failed to detect smoke. In this case, both Yolov5s+ECA and Yolo-Light could detect smoke correctly.

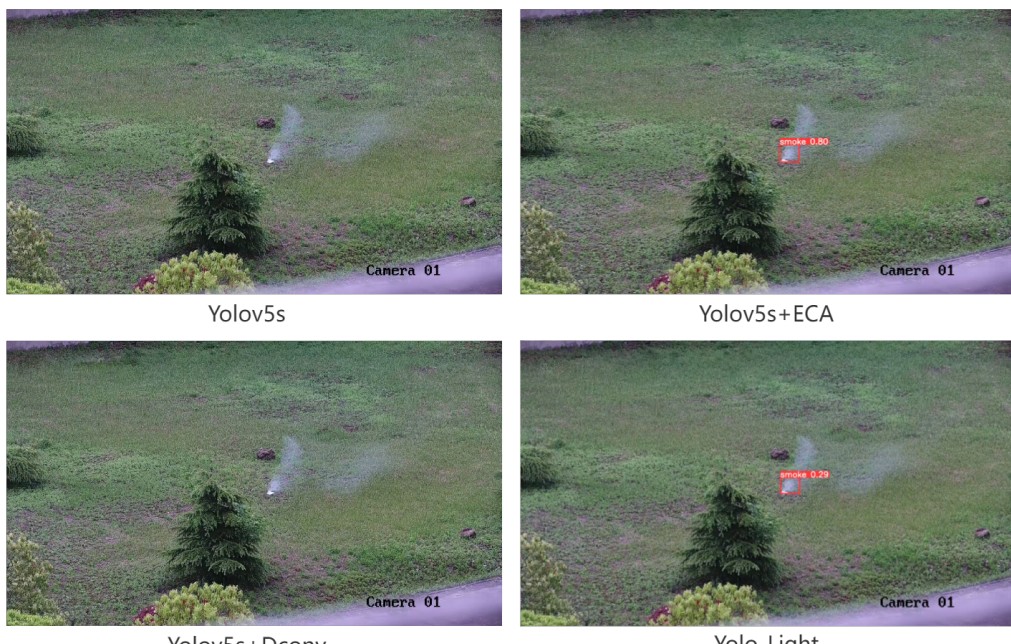

| Yolov5s | Yolov5s+ECA |
|---|---|
| Yolov5s+Dconv | Yolo-Light |

**Figure 11.** Detection results of different algorithms in an unfamiliar scene.

In Figure 12, haze is clearly visible in the scene and the visibility is low. In this case, the algorithm ignores the fuzzy smoke. As a result, both Yolov5s and Yolov5s+ECA were able to detect only one large area of smoke and they missed the smaller area of smoke on the right side of the image; however, Yolov5s+ECA had a higher degree of confidence than Yolov5s. Yolov5s+Dconv did not detect smoke due to computational problems. Yolo-Light successfully detected two smog patches and was unable to handle the blurrier smog in the image.

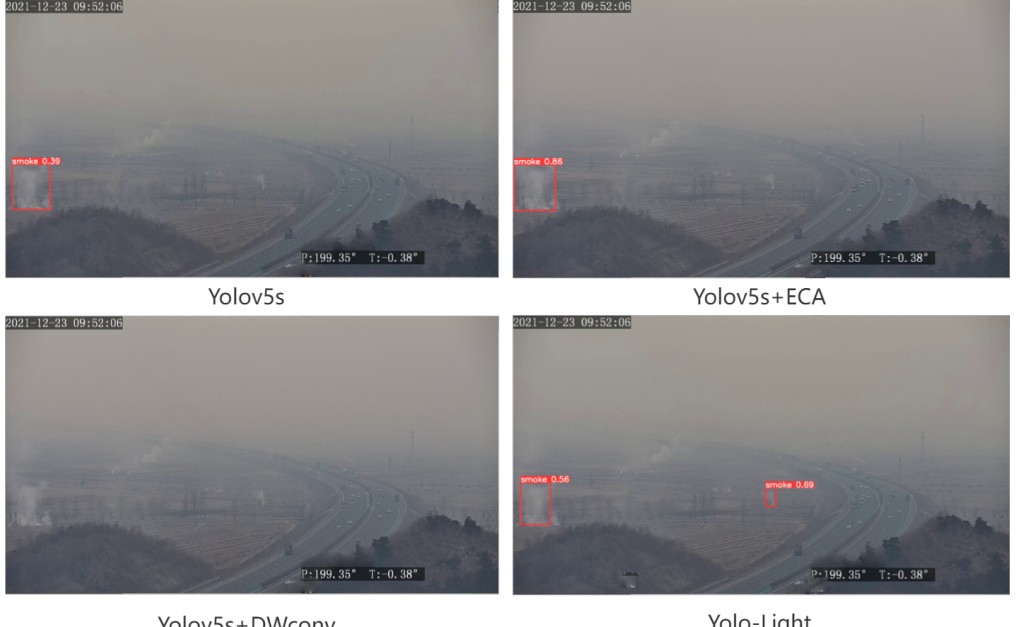

| Yolov5s | Yolov5s+ECA |
|---|---|
| Yolov5s+DWconv | Yolo-Light |

**Figure 12.** Detection results of different algorithms in low visibility scene.

Table 7 shows the comparison results of the actual network performance test. The recognition rate of the network is consistently over 87%, showing that the network has strong generalization ability. The recognition rate of Yolov5s is 90.2%, and that of Yolo-Light is 92.6%.

**Table 7.** Results of model-performance testing.

| Model | Correct Pictures | Recognition Rate |
| --- | --- | --- |
| Yolov5s | 902 | 90.2% |
| Yolov5s+ECA | 907 | 90.7% |
| Yolov5s+Dconv | 876 | 87.6% |
| Yolo-Light | 926 | 92.6% |

Table 8 shows the comparison of Yolo-Light with Yolov5s and Yolov8s (which is the latest version of the Yolo series) [24], further demonstrating the performance of Yolov5s.

**Table 8.** Comparison of Yolo-Light, Yolov5s, and Yolov8s.

| Model | Params | Model Size | mAP.5 | Recognition Rate | FPS on Jetson |
| --- | --- | --- | --- | --- | --- |
| Yolov5s | 7.01 M | 14.4 MB | 0.709 | 90.2% | 22.57 |
| Yolov8s | 11.1 M | 22.5 MB | 0.708 | 89.3% | 15.97 |
| Yolo-Light | 4.76 M | 9.9 MB | 0.750 | 92.6% | 28.65 |

The above experimental results show that the improved Yolo-Light network has a higher detection ability when using the straw-burning smoke target dataset. In terms of actual performance, the average accuracy of Yolo Light is 0.750, the recognition rate on the test set is 92.6%, and the inference speed on NX reaches 28.65 FPS. This indicates that Yolo Light is more suitable for detecting smoke from straw burning compared to Yolov5s and Yolov8s. It is also a small, low-cost, accurate model for use in straw-burning smoke equipment that provides a possible solution. In practical applications, the monitoring camera and Jetson Xavier NX can be mounted in the air at 15 to 20 m, or they can be mounted on the UAV to monitor the straw-burning smoke within 5 km.

## 4. Conclusions

At present, there is no suitable target detection model for detecting smoke from straw burning, and the image background of straw burning is also affected by distance, season, and crop type. In order to explore a recognition algorithm of straw-burning smoke that is suitable for mainstream areas in China, this study designed a two-year collection experiment to enrich the training dataset. Due to the large sizes of the existing target detection models, they are not suitable for deployment on embedded devices with a limited computing performance. Therefore, the Yolov5s target detection network was improved, and deep separable convolution and ECA mechanisms were introduced into the new network model (Yolo-Light) to considerably decrease the number of parameters and calculations. The obtained model volume of the Yolo-Light network was only 4.76 M. Deploying this network on the Jetson Xavier NX could propel the inference speed up to 28.65 FPS.

The straw-burning smoke target detection method proposed in this paper significantly improves the detection speed while considering the recognition rate, and it can be effectively applied to embedded devices with limited computing performance. It also provides a possible solution for miniaturized, low-cost, and rapid straw-burning smoke target detection equipment.

**Author Contributions:** Conceptualization, R.H.; methodology, R.H.; software, C.W.; validation, H.W. (Hao Wang); formal analysis, H.W. (Huanqin Wang); investigation, R.H.; resources, Y.F.; data curation, X.W.; writing—original draft preparation, R.H.; writing—review and editing, H.W. (Huanqin Wang). All authors have read and agreed to the published version of the manuscript.

**Funding:** This work was supported by Major Science and Technology Projects in Anhui Province (NO. 202203a07020004 and NO. 202003a07020005); the National Natural Science Foundation of China (NO. U2133212), and the Natural Science Foundation of Anhui Province (NO. 2208085MF162).

**Institutional Review Board Statement:** Not applicable.

**Informed Consent Statement:** Not applicable.

**Data Availability Statement:** Not applicable.

**Conflicts of Interest:** The authors declare no conflict of interest.

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
