# Peer review of "Yolo-Light: Remote Straw-Burning Smoke Detection Based on Depthwise Separable Convolution and Channel Attention Mechanisms"

_applsci, doi:10.3390/app13095690_

Round 1

Reviewer 1 Report

Comments to the Author:

1. ECA attention mechanism is introduced to improve the SE model in this paper. The depthwise separable convolution is also employed to reduce the parameters. The algorithm has been verified by experiments, which proves the effectiveness of the algorithm proposed in this paper.

2. The author verifies that the illumination conditions in the dataset of the algorithm are relatively good. Can we provide detection results under different illumination conditions, fully demonstrating the robustness of the author's algorithm?

3. The proposed algorithm has been validated using different data sets. Can we further verify the generalization ability of the proposed algorithm based on existing data sets?

Reviewer 2 Report

In this manuscript entitled "Yolo-Light: Remote Straw-Burning Smoke Detection Based on

Depthwise Separable Convolution and Channel Attention Mechanisms" (Manuscript Number: applsci-2314151) I think it’s better to discuss about below questions. Therefore, I suggest a major revision for the manuscript before publication.

Comments:

  1. Abstract: should be start with brief introduction.
  2. Experimental: should be extended. Also, the total materials and companies, characterization of all instruments, methodology and etc. should be presented with more details.
  3. All equations should be referred in the text using a number.
  4. The authors should be explaining about the importance and novelty of the work with more details.
  5. The English language should be improved.
  6. A comparison table should be presented.

Reviewer 3 Report

In this paper, the authors propose a new technique to remotely detect Straw-Burning Smoke utilizing pictures taken in four different locations during the summer and fall seasons of three crops burning  (i.e., wheat, maize, and rice straws). Although the novelty and presentation of the work are moderately poor, I generally think that the paper adds valuable information to monitor straw burning. There are also several instances of typos, grammatical errors, and unclear sentences that need to be revised. Specific comments are mentioned below:

-          INTRODUCTION is missing discussions related to other local pollution primary sources and previous related studies in the area and worldwide.

-          Some sub-sections are randomly divided and named. They require improvement. For example, section 2.2 includes information that can be attached to the previous section.  

-          Most of the written sentences in the result section are superficial. The authors should be deep in their discussion. They should also be selective in adding information and references. 

-          The quality control protocol followed in this study during the collection and modeling process is missing. This is very important for the paper being published.

-          Spell out all the abbreviations at the first use.

-          Line 219: Could you be more specific about the different scenarios?  

Round 2

Reviewer 2 Report

The revised manuscript entitled “Yolo-Light: Remote Straw-Burning Smoke Detection Based on Depthwise Separable Convolution and Channel Attention Mechanisms” (Manuscript Number: applsci-2314151) is rather satisfactory and can be accepted.

Reviewer 3 Report

All the suggested comments have been fully addressed.